# Leading by example: Guiding knowledge transfer with adversarial data augmentation

**Arne Nix[1-2,*], Max F. Burg[2-3], Fabian H. Sinz[1-2,**]**

[1] Institute for Bioinformatics and Medical Informatics, University of Tübingen
[2] Campus Institute Data Science, University of Göttingen
[3] Institute for Theoretical Physics, University of Tübingen

[*]`arne.nix@uni-goettingen.de`, [**]`sinz@cs.uni-goettingen.de`

## Abstract

Knowledge distillation (KD) is a simple and successful method to transfer knowledge from a teacher to a student model solely based on functional activity. However, it has recently been shown that this method is unable to transfer simple inductive biases like shift equivariance. To extend existing functional transfer methods like KD, we propose a general data augmentation framework that generates synthetic data points where the teacher and the student disagree. We generate new input data through a learned distribution of spatial transformations of the original images. Through these synthetic inputs, our augmentation framework solves the problem of transferring simple equivariances with KD, leading to better generalization. Additionally, we generate new data points with a fine-tuned Very Deep Variational Autoencoder model allowing for more abstract augmentations. Our learned augmentations significantly improve KD performance, even when compared to classical data augmentations. In addition, the augmented inputs are interpretable and offer a unique insight into the properties that are transferred to the student.

## 1 Introduction

Knowledge distillation (KD) [Hinton and Dean, 2015] and other functional transfer methods [McClure and Kriegeskorte, 2016, Zagoruyko and Komodakis, 2017] are powerful and flexible tools to transfer the knowledge of a given *"teacher"* model to the transfer target, the *"student"* model, without copying the weights. Instead, these methods match the student's functional activity (e.g. the softmax output in the case of KD) to that of the teacher for presented inputs. This makes them independent of architectural details and allows functional transfer methods to be applied in scenarios like model compression [Bucilă et al., 2006, Hinton and Dean, 2015], continual learning [Pan et al., 2020, Titsias et al., 2019, Benjamin et al., 2019] or even neuroscience [Li et al., 2019], where traditional transfer learning would be impossible to use. Functional transfer methods also appear to be key to training new models that trade off inductive biases for more flexibility and more parameters [Vaswani et al., 2017, Dosovitskiy et al., 2020, Tolstikhin et al., 2021] on smaller data [Touvron et al., 2020, Chen et al., 2022, Nix et al., 2022]. Recently, Nix et al. [2022] showed that current functional transfer methods fail to transfer even simple equivariances between teacher and student. We hypothesize, however, that functional transfer methods are in principle capable of transferring most knowledge from teacher to student if the training data is chosen adequately. We confirm this hypothesis on a small toy example (Section 2), showing the importance of input data for functional transfer. Motivated by this, we propose a general framework (Section 3) to generate synthetic training inputs which improve knowledge transfer by maximizing the disagreement between teacher and student while leaving the teacher's output unchanged. Consequently, our framework moves the input in directions that the teacher is invariant to but which are most challenging for the student. In our experiments

NeurIPS 2022 Workshop on Synthetic Data for Empowering ML Research.

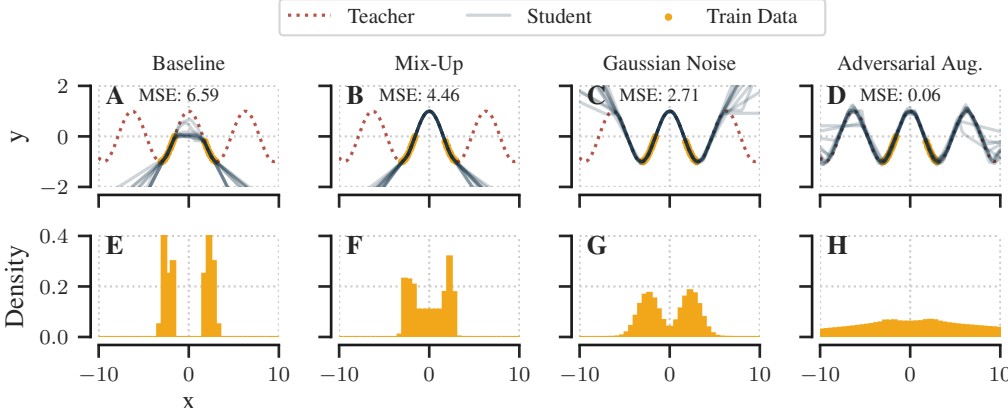

Figure 1: Fitting the student, a ReLU network, to the teacher function, $\cos(x)$ for 10000 iterations. We show the resulting models for 10 random seeds (**A-D**) and the distribution of (augmented) training inputs as a normalized histogram (**E-H**). We compare Baseline (no augmentations) with Mix-Up, Gaussian Noise and our Adversarial Augmentations. Mean-squared-error (MSE) reported on 100 test inputs sampled from $\mathcal{U}_{[-10,10]}$.

(Section 4) we show that our task-agnostic framework improves the effectiveness of the transfer and thereby solves the problem of inductive bias transfer for the example of shift equivariance proposed by Nix et al. [2022]. Additionally, we formulated the synthetic data generator as a parameterized augmentation (Section 3.1) that can be integrated with most existing functional transfer methods. Finally, we demonstrate that our framework learns interpretable augmentations that improve KD to the same level and in some cases even beyond established data augmentation methods.

**Related Work** Our method is inspired by the large body of work investigating different types of data augmentations: Golan et al. [2020] optimize new data points that lie by construction at the decision boundary of two models, i.e. at a point at which the decisions of both models do not coincide anymore. Many studies use parameterized augmentations optimized to improve a given objective [Cubuk et al., 2018, Rusak et al., 2020, Hendrycks et al., 2021, Zietlow et al., 2022], and some even optimize the augmentations to improve on an adversarial objective [Volpi et al., 2018, Behpour et al., 2019, Zhang et al., 2019a,b, Zhao et al., 2020, Gong et al., 2021, Antoniou et al., 2022], however, without applying them for knowledge transfer. In KD, applying data augmentations is a very effective tool to improve matching between student and teacher [Wang et al., 2020, Beyer et al., 2021] and optimizing on a meta level can be useful to aide the teaching [Pham et al., 2021]. Similar to our work, Rashid et al. [2021], Haidar et al. [2022], Zhang et al. [2022] utilized adversarial objectives to optimize data augmentations for KD, however, they were solely focused on natural language processing tasks and do not optimize the augmentations towards invariance.

Inspired by this large body of work we formulate a task-agnostic framework containing only one task specific building block – the instantiation of the augmentor – for which we offer the reasonable model choice of a spatial transformer [Jaderberg et al., 2015] or a very deep variational autoencoder [Child, 2020, VDVAE] and successfully apply our framework to image classification.

## 2 Input Data Matters for Functional Transfer

We hypothesize that the choice of input data is crucial to successfully transfer knowledge based on functional activity alone. To demonstrate this, we consider a simple KD task with the true function $\cos(x)$ as the teacher and a small three layer neural network with ReLU activation [Agarap, 2018] as the student. Assume that the model input is $\mathcal{X}_{\text{train}} \subset (-\pi, -\frac{\pi}{2}) \cup (\frac{\pi}{2}, \pi)$. In this scenario the student would not capture the teacher's periodicity failing to extrapolate beyond the training data (Figure 1A). Picking "better" training inputs would obviously mitigate this problem, but it is unclear how to select these, especially in high dimensions where the search space quickly becomes infeasibly large.

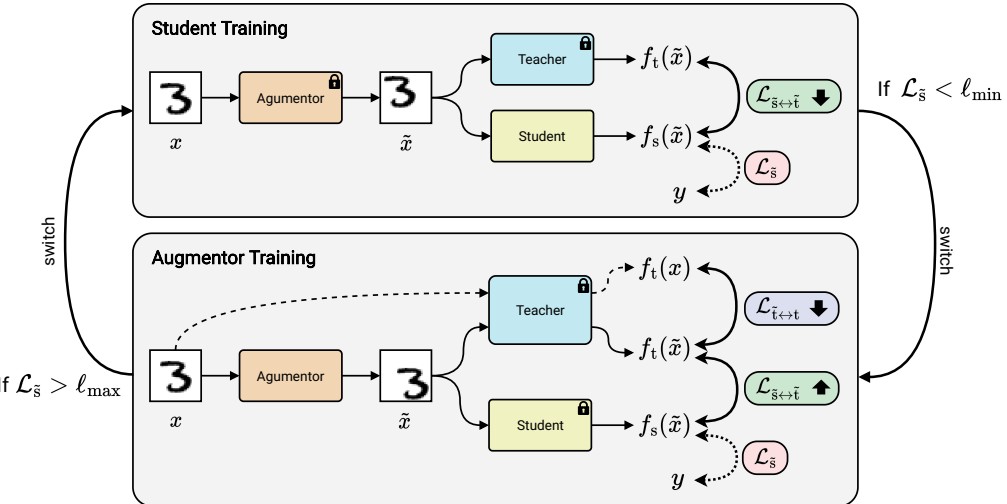

Figure 2: Our task-agnostic framework that switches between training the student and training the augmentor.

To improve KD, Beyer et al. [2021] suggest to use mixup [Zhang et al., 2017] and data augmentations. Using mixup in our 1D example would be equivalent to interpolating between pairs of input points, $\tilde{x} = (1-\alpha)x_1 + \alpha x_2$ for $\alpha \sim \mathcal{U}_{[0,1)}$. Mixup clearly allows the student to interpolate between training points, but does not enhance extrapolation (Figure 1B). In our 1D example, we augment data by adding noise from a standard normal distribution, $\tilde{x} = x + \epsilon$ with $\epsilon \sim \mathcal{N}(\mu = 0, \sigma = 1)$. This helps our student to match the teacher function beyond the original training regime (Figure 1C). However, the student only improves within a fixed margin that is determined by mean $\mu$ and standard deviation $\sigma$ of the noise distribution. While lifting this constraint by heuristically or randomly exploring different $\mu$ and $\sigma$ seems to be an easy solution in our 1D example, it is unclear how to select these in the general case of high-dimensional inputs with a search-space that cannot be tractably explored at random. Instead, we propose to optimize a parameterized augmentation (in this 1-D example we could for instance learn $\mu$ and $\sigma$) to generate synthetic datapoints such that we can maximize the learning benefit for our student, matching the teacher on a much larger area (Figure 1D,H).

## 3  Adversarial Augmentations

**Teacher-Student loss**  In KD and most other functional transfer methods the objective is to minimize a distance $\mathcal{D}\left[f_s(x), f_t(x)\right]$ between the student's activation $f_s(x)$ and the teacher's activation $f_t(x)$ on given inputs $x \in \mathbb{R}^n$. In the case of KD, $\mathcal{D}$ would be the Kullback-Leibler divergence between the softmax distributions of teacher and student. Unfortunately, minimizing this objective on the training data only can miss properties of the teacher that are crucial for generalization (Figure 1A).

Our goal is to allow the transfer of such properties by generating data points (or augmentations) $\tilde{x} = g_a(x)$ that help the student learn from the teacher. Hence, we consider the more general case of matching student and teacher on augmented inputs $\tilde{x} \in \mathbb{R}^n$:

$$\mathcal{L}_{\tilde{s}\leftrightarrow\tilde{t}} = \mathcal{D}\left[f_s(\tilde{x}), f_t(\tilde{x})\right]. \tag{1}$$

The student's parameters $\theta_s$ are optimized as usual to minimize this objective. However, the augmentor $g_a$ and its parameters $\theta_a$ are trained to maximize the same objective that the student is trying to minimize. This leads us to an adversarial framework [Goodfellow et al., 2014] for augmentation.

**Teacher-Teacher loss**  We further propose a second loss to lead the augmentor towards generating data points to which the teacher is invariant, as these are often useful directions for generalization. This can be achieved by minimizing the distance between the teacher's output for the original data $x$ and its output for the augmented data $\tilde{x}$:

$$\mathcal{L}_{\tilde{t}\leftrightarrow t} = \mathcal{D}\left[f_t(\tilde{x}), f_t(x)\right] \tag{2}$$

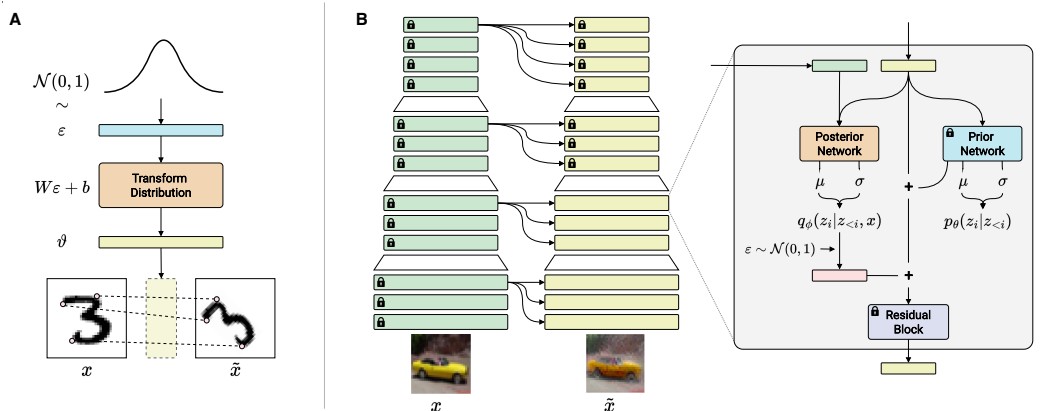

Figure 3: Illustration of the augmentor models used in our experiments. **(A)** Learned distribution of affine transformations in the pixel coordinates. **(B)** Finetuning the posterior of a trained VDVAE.

**The Algorithm**    To summarize, we optimize the augmentor's parameters $\theta_a$ to generate augmentations difficult for the student and simultaneously optimize the student's parameters to perform well on those augmentations.

$$\max_{\theta_a} \; \lambda_s \mathcal{L}_{\tilde{s} \leftrightarrow \tilde{t}} - \lambda_t \mathcal{L}_{\tilde{t} \leftrightarrow t} \qquad \text{and} \qquad \min_{\theta_s} \; \mathcal{L}_{\tilde{s} \leftrightarrow \tilde{t}} \qquad (3)$$

We further propose to simplify optimization by separating student and augmentor training (Figure 2), which is possible because $\mathcal{L}_{\tilde{t} \leftrightarrow t}$ prevents a diverging solution for our augmentor. We switch from training the student to training the augmentor if the student's performance $\mathcal{L}_{\tilde{s}}$ on augmented data surpasses a pre-defined threshold $\ell_{\min}$. Analogously, we switch back from augmentor to student training, if $\mathcal{L}_{\tilde{s}}$ exceeds $\ell_{\max}$, i.e. if the augmentations are hard enough. Continuing this cycle leads to a task-agnostic, general algorithm that trains both augmentor and student.

## 3.1 The Augmentor Models

We generate new input data points using an *augmentor* model $g_a$. It is important to choose an augmentor that fits the desired application and that is powerful enough to generate helpful augmentations. We usually do not know a priori what useful augmentations are and thus should try to allow as much flexibility as possible. While our framework is universally applicable, choosing an effective augmentation model is a big challenge that likely needs to be addressed for each task separately.

**Spatial Transformer**    As a first, model to augment image data, we propose the use of a simple *spatial transformer* module [Jaderberg et al., 2015, ST]. Spatial transformers are parameterized modules for manipulations of an input image. For our augmentor, we learn a distribution over the entries of an affine transformation matrix $\vartheta \in \mathbb{R}^{2 \times 3}$ that defines the transformation of the sampling grid, i.e. a transformation of the positions that pixels from the original image are mapped to in the augmented one (Figure 3A). This augmentor has only 42 learnable parameters (one fully-connected layer), but allows for augmentations such as cropping, translation, rotation, scale, and skew.

**Very Deep VAE**    In order to enable augmentations that go beyond spatial manipulations of a given image, we propose the use of a generative image model as the augmentor. This would allow us to potentially explore the entire image space to generate synthetic data, while conditioning on our original training samples. We chose the *very deep VAE* model [Child, 2020, VDVAE] as our augmentor model, as this is a powerful generative model with nearly perfect reconstructions and near state-of-the-art likelihood scores on established benchmarks. Furthermore, this augmentor can generate images quickly in a single forward pass and its hierarchical structure and small latent dimensions allow for easy manipulation of image components in different levels of detail.

We use a pre-trained VDVAE that we finetune with our objectives to produce the desired augmented version of the original image instead of a perfect reconstruction. To do so, we leverage the hierarchical structure of the VDVAE and finetune solely the parameters of the posterior network from layer 10

Table 1: MNIST (columns "Centered") and MNIST-C (columns "Shifted") test accuracies (mean and standard error of the mean across 4 random seeds) comparing KD without augmentation and our adversarial augmentations with Orbit transfer. Left two columns show the transfer results from a small CNN teacher to an MLP student. The right columns show analogous experiments between a ResNet18 teacher and a small ViT student. The best performing transfer is shown in bold for each column. Examples of our learned data augmentations shown on the right. "Adv Aug (*Shifted*)" shows best-performing settings for the shifted test set (i.e. not selected based on Validation).

| Method | CNN → MLP | | ResNet18 → ViT | |
| --- | --- | --- | --- | --- |
| | Centered | Shifted | Centered | Shifted |
| Teacher only | $99.0 \pm 0.0$ | $91.3 \pm 0.5$ | $99.5 \pm 0.0$ | $92.8 \pm 0.5$ |
| Student only | $98.4 \pm 0.0$ | $35.2 \pm 0.7$ | $98.3 \pm 0.0$ | $40.4 \pm 0.8$ |
| + Random ST | $92.1 \pm 0.6$ | $81.0 \pm 2.0$ | $95.4 \pm 0.3$ | $90.4 \pm 1.0$ |
| + Shifts | $98.1 \pm 0.1$ | $86.5 \pm 0.3$ | $98.5 \pm 0.0$ | $93.7 \pm 0.2$ |
| Orbit [Nix et al., 2022] | **98.8** | **95.2** | 98.4 | **84.0** |
| KD | $98.6 \pm 0.0$ | $40.3 \pm 0.6$ | $98.6 \pm 0.1$ | $44.7 \pm 1.9$ |
| + AdvAug (Validation) | $98.6 \pm 0.1$ | $68.9 \pm 2.5$ | **99.2** $\pm 0.0$ | **84.1** $\pm 2.3$ |
| + AdvAug *(Shifted)* | *98.5 $\pm$ 0.1* | *91.7 $\pm$ 0.4* | *99.2 $\pm$ 0.0* | *89.3 $\pm$ 0.8* |

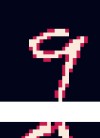
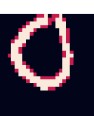
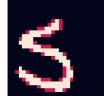

onward in the decoder (Figure 3B). The encoder, decoder layers closer to the input, as well as prior and res-block in the selected layers are frozen and remain unchanged from the pre-trained state.

# 4   Experiments

**Transferring Shift Equivariance**   For our initial experiment, we reproduce the setup from Nix et al. [2022] to test whether we can transfer the inductive bias from a shift equivariant teacher, CNN and ResNet18 [He et al., 2015], to a student that does not have this inductive bias built into its architecture: a Multi-Layer Perceptron (MLP) and a Vision Transformer (ViT). When training the students and teachers by themselves on standard MNIST [Deng, 2012] training data, we observe a small difference in generalization performance (-0.6% and -1.2%) between teacher and student on the MNIST test set and a large gap (-56.1% and -52.4%) when we evaluate on a version of the test set in which digits were randomly shifted [Mu and Gilmer, 2019]. As another baseline, we applied plain KD in an attempt to transfer shift equivariance from teacher to student. Consistent with the findings of Nix et al. [2022], we only observe a small improvement on the centered (+0.2% and +0.3%) and the shifted (+5.1% and +4.3%) test sets, which is likely caused by the centered training data we use for transfer.

We then test if combining KD with our augmentations produced by a spatial transformer augmentor would outperform these baselines. The resulting student model shows significantly better performance on shifted inputs (+28.6% and +39.4%) compared to plain KD and the generated images clearly show that the augmentor learns to shift the digits within the image. To verify the potential performance ceiling of our method, we additionally report results for hyper-parameter settings (different learning-rate) that were chosen based on performance on the shifted test data instead of the centered validation set. This shows us that we could potentially get very close to the teacher's performance on this metric. Compared to Nix et al. [2022] our approach outperforms their results on the VIT task and gets close to their result with an MLP student when we use our best hyper-parameter configuration. This demonstrates that our method while acting on fewer parts of the network (only on the input of the student) and while being a more general method (can be added to most existing functional transfer methods), can reach similar or even better performance when it comes to transferring invariances.

We perform two control experiments to verify that the student's performance improvement is specifically due to our data generation framework. The first experiment (Random ST) augments the training inputs of a stand-alone student model with a spatial transformer that performs input augmentations using parameters sampled within a reasonable range (i.e. ensuring the digit is always fully visible). Although the student performs well evaluated on the shifted test set, performance degrades a lot on the centered test set. Our spatial transformer augmentation, despite being trained without any bound on the transformations, learns more useful augmentations than the ones of this control experiment.

Table 2: Test accuracies on the CIFAR10 test set. Standard error is reported where available across three different seeds. Best transfer is highlighted in bold. The ResNet101* models were pretrained on ImageNet. Example augmentations of test images from ResNet18→ViT experiments with samples across different iterations.

| | ResNet18 ↓ ViT | ResNet101* ↓ ViT | ResNet101* ↓ ResNet18 |
|---|---|---|---|
| Teacher only | $92.5 \pm 0.0$ | 95.5 | 95.5 |
| Student only | $68.5 \pm 0.5$ | 69.4 | 78.5 |
|   + Standard Aug. | $78.3 \pm 0.4$ | 78.1 | 92.6 |
|   + Random ST Aug. | $58.9 \pm 0.4$ | 58.6 | 79.3 |
| KD | $67.9 \pm 0.1$ | 68.5 | 84.4 |
|   + Standard Aug. | $80.9 \pm 0.1$ | 79.3 | 93.3 |
|   + Adv. Aug. (ST) | $\mathbf{87.8 \pm 0.8}$ | 84.4 | 93.5 |
|   + Adv. Aug. (VDVAE) | $81.9 \pm 0.4$ | 81.2 | 91.0 |
|   + Adv. Aug. (VDVAE + ST) | $87.6 \pm 0.6$ | $\mathbf{87.1}$ | $\mathbf{94.0}$ |

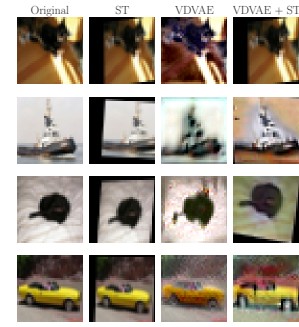

In our second control (Shifts) we asked how much data augmentations could improve the performance in the best case (without KD). For this, we just augment the inputs with the same random shifts that are applied to obtain the shifted test data. We see that training with these augmentations leads to great improvements on the shifted test set. Nevertheless, our learned augmentations achieve scores in a similar range on the shifted evaluation and outperform its results on the centered test set.

**Transfer on Natural Images**  To test our framework on a more realistic problem, we decided to apply it to CIFAR10 [Krizhevsky et al.] on three different transfer scenarios (see Table 2). This reaches from experiments where the student lacks an inductive bias (ResNet18→ViT), over instances where the teacher has more capacity and access to data than the student (ResNet101* → ResNet18), to settings where both are the case (ResNet101* →ViT). We attempted to keep the experimental setup as close to the previous MNIST experiments as possible (see Appendix A for details).

First, we observe that inductive biases and data augmentations play an important role on the relatively small task of CIFAR10. A ViT without knowledge distillation performs quite poorly, reaching only 78.3% accuracy. The performance is even worse (68.5%) when it is trained without the standard set of data augmentations consisting of random rotations, cropping, and horizontal flips. That these are generally a crucial component of training is also evident with even CNN-based student models losing 14.1% and 17.4% when trained without these augmentations. This makes it particularly noteworthy that the data augmentations that were learned by our ST augmentor can surpass the performance of these well-established augmentations for KD. Especially for the ViT student that does lack strong inductive biases compared to its teacher, we see a significant improvement compared to KD with standard augmentations. Qualitatively, the data augmentations produced by our augmentor demonstrate a large variety of spatial transformations, clearly showing that a difference in performance on these examples is what separates student and teacher the most.

We also see significant improvement over the standard data augmentations when employing the VDVAE augmentor for transfer to a ViT student. However, looking at the resulting images indicates that our augmentor lacks the expected shifts of object positions, but rather learns stylistic changes in the image. This finding motivated us to combine our two augmentors (VDVAE + ST). The resulting images demonstrate variability in both style and spatial alignment, leading to the best performance for all teacher-student pairings.

## 5   Conclusion

In this work we introduced a general, task-agnostic and modular framework to extend functional transfer methods by parameterized data augmentations. The augmentation models are optimized to generate inputs on which teacher and student disagree, keeping the teacher's predictions unchanged at the same time. We show that these augmentations can solve the issue of knowledge transfer even if the teacher's inductive biases are distinct from the student's. We further demonstrate that our learned augmentations achieve performance competitive to established classical data augmentation

techniques even when student and teacher share similar inductive biases. Overall our framework offers a promising tool that enhances transfer performance and offers a unique window into the transferred knowledge through its interpretable augmentations.

**Acknowledgements**

We thank all reviewers for their constructive and thoughtful feedback. Furthermore, we thank Mohammad Bashiri, Pawel Pierzchlewicz and Suhas Shrinivasan for helpful comments and discussions. The authors thank the International Max Planck Research School for Intelligent Systems (IMPRS-IS) for supporting Arne Nix and Max F. Burg.

This work was supported by the Cyber Valley Research Fund (CyVy-RF-2019-01), by the German Federal Ministry of Education and Research (BMBF) through the Tübingen AI Center (FKZ: 01IS18039A), by the Deutsche Forschungsgemeinschaft (DFG) in the SFB 1233, Robust Vision: Inference Principles and Neural Mechanisms (TP12), project number: 276693517, and funded by the Deutsche Forschungsgemeinschaft (DFG, German Research Foundation) – Project-ID 432680300 – SFB 1456. FHS is supported by the Carl-Zeiss-Stiftung and acknowledges the support of the DFG Cluster of Excellence "Machine Learning – New Perspectives for Science", EXC 2064/1, project number 390727645.

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

# A   Setup Details

**Training**    We train on the entire CIFAR10 dataset (excluding 10% held-out as validation set) for 300 epochs with a batch-size of 256. As an optimizer, we use Adam [Kingma and Ba, 2014] with a learning rate of 0.0003 and an L2-regularization of $2 \cdot 10^{-9}$. Our training begins with a linear warmup of the learning rate for 20 epochs. The validation accuracy is monitored after every epoch and if it has not improved for 20 consecutive epochs, we decay the learning rate by a factor of 0.8 and restore the previously best performing model. The training is stopped prematurely if we decay five times.

**Models**    The different models we use generally follow the standard architecture and settings know from the literature. For the ViT, we use a smaller variant of it on the CIFAR task. It consists of six layers and eight attention heads throughout the network. The dropout rate is set to 0.1 and the hidden dimension is chosen as 512 in all places.

**KD and Adversarial Augmentations**    After initial experiments on MNIST, we decided to use a softmax temperature of 5.0 for all experiments involving KD. We furthermore rely solely on the KL-Divergence loss to optimize our model. For the experiments with our augmentation framework, we have the same settings as before for the student (KD) training and separate settings for the augmentor training. There we reduce the batch-size to 160 (128) and a learning-rate of 0.0001 (0.05) for the VDVAE augmentor (ST augmentor). We initialize both augmentors to perform an identity transformation, i.e. the VDVAE is taken pretrained from Child [2020]. The thresholds for switching are set as $\ell_{\min} = 10\%$ (5%) and $\ell_{\max} = 60\%$ (40%). The train modi are switched if the threshold is surpassed for 5 consecutive iterations. Both $\lambda_s$ and $\lambda_t$ are set to 1 for the experiments. For the experiment ResNet101$^*$ $\rightarrow$ ResNet18, we found a slightly different setting to be more effective with $\ell_{\min} = 5\%$ and $\ell_{\max} = 40\%$ and a switch only happening if the threshold is surpassed for 10 consecutive iterations.

