# OpenReview forum: "Leading by example: Guiding knowledge transfer with adversarial data augmentation"
_NeurIPS.cc/2022/Workshop/SyntheticData4ML — Neurips 2022 SyntheticData4ML_

### Official Review · Reviewer_oaLQ · 2022-10-18
**A well-written paper but claimed generality of the approach is not supported by experiments beyond CV tasks**

**Rating:** 6
**Confidence:** 4

**Review:**

- Authors propose a method for knowledge distillation by generating adversarial examples where student and teacher disagree. To this end, they train augmentor to maximise the difficulty for the student while minimizing teacher predictions between the original and augmented sample. The idea is that this allows finding points that are outside the range covered in the training set but to which the teacher is invariant.
- Authors compare their approach to several baselines both in the section with the toy example and for the two CV tasks.
- The paper is well-written and is grounded in the literature
- Authors claim the generality of their approach but only test on computer vision tasks. CV is notoriously responsive to data augmentation approaches. Thus, it would be interesting to see whether the proposed approach works well in other domains, for example, in different NLP tasks.
    - One point that is not discussed is how the proposed augmentations and training regime affect the input distribution that the student sees. In many real-world tasks preserving the original input distribution of the training set is vital for good performance.
- Doesn’t look like the code is available to reproduce the experiments.
- Typos:
    - Nix et al. 2022 reference has some broken links: the link to the paper got mixed up with the link to the source code and the repository got moved to https://github.com/sinzlab/orbit_transfer

---

### Official Review · Reviewer_AmUA · 2022-10-18
**Good paper**

**Rating:** 7
**Confidence:** 3

**Review:**

This paper introduces a way to generate examples of knowledge distillation by a VDVAE. The paper is well written and the experimental results suggest the generated examples improved the student model compared with some baseline image perturbations. Here are some comments for this paper.

1. An ablation is missing for using **non finetuned VDVAE** model for the data generator of the student model training.

2. It would be better to show if the whole training method is robust to the hyperparameters.

3. It would be better to show how much extra computation is needed when using a VDVAE model for the distillation.

---

### Meta-Review · Area_Chair_vELP · 2022-10-19

**Recommendation:** Accept